# Development of a recombinase polymerase amplification assay with lateral flow dipstick (RPA-LFD) for rapid detection of *Shigella* spp. and enteroinvasive *Escherichia coli*

Zheng Bian[1], Wenbo Liu[1], Junhua Jin[1], Yanling Hao[2], Linshu Jiang[3], Yuanhong Xie[1], Hongxing Zhang[1]*

1 Beijing Laboratory of Food Quality and Safety, Beijing Key Laboratory of Agricultural Product Detection and Control of Spoilage Organisms and Pesticide Residue, Beijing Engineering Technology Research Center of Food Safety Immune Rapid Detection, College of Food Science and Engineering, Beijing University of Agriculture, Beijing, China, 2 Department of Nutrition and Health, Key Laboratory of Functional Dairy, Co-constructed by Ministry of Education and Beijing Government, China Agricultural University, Beijing, China, 3 Animal Science and Technology College, Beijing University of Agriculture, Beijing, China

* hxzhang51@163.com

**Data Availability Statement:** All relevant data are within the paper and its Supporting information files.

## Abstract

*Shigella* spp. and enteroinvasive *Escherichia coli* (EIEC) are widely distributed and can cause serious food-borne diseases for humans such as dysentery. Therefore, an efficient detection platform is needed to detect *Shigella* and EIEC quickly and sensitively. In this study, a method called recombinase polymerase amplification combined with lateral flow dipstick (RPA-LFD) was developed for rapid detection of *Shigella* and EIEC. RPA primers and LFD detection probes were designed for their shared virulence gene *ipaH*. Primers and probes were screened, and the primer concentration, and reaction time and temperature were optimized. According to the optimization results, the RPA reaction should be performed at 39˚C, and when combined with LFD, it takes less than 25 min for detection with the naked eye. The developed RPA-LFD method specifically targets gene *ipaH* and has no cross-reactivity with other common food-borne pathogens. In addition, the minimum detection limit of RPA-LFD is $1.29 \times 10^2$ copies/µL. The detection of food sample showed that the RPA-LFD method was also verified for the detection of actual samples.

## Introduction

*Shigella* spp. and enteroinvasive *Escherichia coli* (EIEC) are highly important human pathogens. More than a million deaths and 160 million cases are attributed to shigellosis every year worldwide. *Shigella* spp. and EIEC pose a serious threat to many countries, responsible for the majority of cases of endemic bacillary dysentery prevalent in developing countries [1–3]. Studies have shown that *Shigella* can contaminate many foods, including vegetables, meat, egg products, and dairy products, and EIEC is also present in a variety of foods [4, 5]. The conventional national standard method is bacterial enrichment and selective cultivation, followed by

**Funding:** We thank the Research project of Beijing Municipal Commission of Education, this work was supported by the Research project of Beijing Municipal Commission of Education (KM201810020016). The funders had no role in study design, data collection and analysis, decision to publish, or preparation of the manuscript.

**Competing interests:** The authors have declared that no competing interests exist.

biochemical and serological methods for detection and typing [6]; however, this process is lengthy and often takes several days, which makes detection efficiency extremely low [7]. Since the invasion-related gene *ipaH* is present in both *Shigella* and EIEC, simultaneous detection of both bacteria is possible. Therefore, a fast, specific, and sensitive method is needed to detect *ipaH* in the food processing and clinical contexts, as it will greatly improve public health.

In recent years, several polymerase chain reaction (PCR)-based detection techniques for *Shigella* have been developed. These methods are more rapid, accurate, specific, sensitive, and stable than the methods described above [8, 9]. However, because they involve bulky equipment, time-consuming thermal cycling steps, and require trained professionals, their usefulness is limited in resource-poor environments [10]. In order to avoid the need for thermal cycling equipment for amplification, isothermal nucleic acid amplification methods have been developed in laboratories. These tests mainly include nucleic acid sequence-based amplification (NASBA) [11], strand displacement amplification (SDA) [12], loop-mediated amplification (LAMP) [13], rolling circle amplification (RCA) [14], helicase-dependent amplification (HDA) [15], and recombinase polymerase amplification (RPA) [16]. As a new type of isothermal amplification technology, RPA only needs two primers to realize a reaction. The temperature required for the reaction is 37–42˚C, i.e., the amplification can occur at room temperature, and detection can be achieved within 20 min.

At present, commonly used RPA product detection methods mainly include agarose gel electrophoresis [16], real-time fluorescence [17], chemical color development [18], electrochemistry [19], and lateral flow test strips (LFD) [20]. LFD is an extremely simple and fast detection method, which usually only takes 2–5 min to produce results; additionally, small size and easy storage make it suitable for rapid on-site detection. In this study, we describe a detection method involving RPA and LFD. We designed specific primers and probes for the *ipaH* gene for the rapid detection of *Shigella* and EIEC. The sensitivity and specificity of our detection method were also evaluated. The sensitivity was compared to that of a method involving RPA and agarose gel electrophoresis. At the same time, the application of this method in food sample detection was evaluated.

## Materials and methods

### Bacterial strains and DNA extraction

The strains used in this study are listed in Table 1. All strains were provided by our laboratory (Beijing Laboratory of Food Quality and Safety, Beijing University of Agriculture, Beijing, China). *Shigella flexneri* ATCC 12022 was used to determine optimal conditions for RPA-LFD assays. Bacterial genomic DNA was extracted using the TIANamp Bacteria DNA Kit (TIANGEN BIOTECH Co., Ltd, Beijing, China) according the manufacturer's instructions, and the extracted DNA was stored at -20˚C until use.

### Design and optimization of RPA primers

RPA primers (S1 Table) specific for the *ipaH* gene were designed using the TwistAmp® reaction kit manual (TwistDx, Cambridge, UK). The highly conserved nucleotide sequence of *ipaH* (NCBI reference sequence: NC_004337.2) was selected as the target sequence. According to the principle of RPA primer design, primers were designed and screened using the primer-BLAST function of the National Center for Biotechnology Information (NCBI). The primers were screened according to the TwistAmp® Basic Quick Guide using the TwistAmp® Basic kit (TwistDx, Cambridge, UK), in a 50 μL reaction system, including 29.5 μL Buffer, 1 μL of the extracted DNA, 12.2 μL ddH$_2$O, and 0.48 μM of each of the forward and reverse primers. After mixing, 2.5 μL of Mg(Ac)$_2$ (280mM) was added to start the reaction. After the reaction,

**Table 1. Bacterial strains used in the study.**

| Strain name | Strain code |
| --- | --- |
| *Shigella flexneri* | ATCC[a] 12022 |
| *Shigella sonnei* | ATCC 25931 |
| *Shigella boydii* | ATCC 9207 |
| *Shigella dysenteriae* | ATCC 13313 |
| *Salmonella* Typhimurium | ATCC 50220 |
| *Listeria monocytogenes* | ATCC 19111 |
| *Staphylococcus aureus* | ATCC 25923 |
| *Campylobacter jejuni* | ATCC 29428 |
| *Pseudomonas aeruginosa* | ATCC 9027 |
| *Clostridium perfringens* | ATCC 13124 |
| *Vibrio parahaemolyticus* | ATCC 17802 |
| *Enteroinvasive E. coli* | ATCC 43893 |
| *Enteroinvasive E. coli* | CICC[b] 10661 |
| *Enteroinvasive E. coli* | CICC[b] 10662 |
| *Enteroinvasive E. coli* | CICC[c] 24188 |
| *Enterotoxigenic E. coli* | ATCC 35401 |
| *Enteroaggregative E. coli* | ATCC 9610 |
| *Shiga toxin-producing E. coli* | ATCC 43895 |
| *Enteropathogenic E. coli* | ATCC 43887 |
| *Enterohemorrhagic E. coli* | ATCC 35150 |

[a] ATCC, American Type Culture Collection.

[b] CICC, China Center of Industrial Culture Collection, Source: Shanghai Center for Disease Control and Prevention.

[c] Source: Henan Entry Exit Inspection and Quarantine Bureau.

the solution was purified by the phenol-chloroform method [21]. The purified RPA product was analyzed by 2% agarose gel electrophoresis, then stained with 4S Red Plus Nucleic Acid Stain, and observed with a WD-9413B imaging analyzer (Beijing Liuyi Biotechnology Co., Ltd., Beijing, China).

## Design and optimization of RPA-LFD probe

The probes and primers (S2 Table) used in the RPA-LFD reaction were prepared in accordance with the following principles: 5-Carboxyfluorescein (FAM) at the 5' end of the probe, C3-spacer at the 3' end, Tetrahydrofuran (THF) in the middle, and biotin at the 5' end of the reverse primer. All primers and probes were synthesized by Sangon Biotech Co., Ltd (Shanghai, China).

## RPA-LFD procedure

The RPA-LFD was performed using the RAA-nfo kit as per the manufacturer's instructions (Hangzhou ZC Bio-Sci & Tech Co. Ltd, Hangzhou, China), in a 50 μL reaction system, including 37.9 μL A Buffer, 0.024 μM probe, 5 μL of the extracted DNA, 2.5 μL B Buffer, and 0.08 μM of each of the forward and reverse primers. After 30 min of incubation at 39°C, the RPA product was purified and transferred to a clean PCR tube. Next, the binding pad end of the LFD (Universal lateral flow strips; rainbow) (Tiosbio BIOTECH Co., Ltd, Beijing, China) was inserted into the PCR tube, and the result was interpreted once the control line was present.

### Optimization of RPA-LFD primer concentration

In order to determine the effect of primer concentration on the reaction, different final concentrations of primers were used for the RPA-LFD reaction: 0.08 μM, 0.16 μM, 0.24 μM, 0.32 μM, and 0.40 μM.

### Optimization of RPA-LFD reaction temperature

The reaction temperature was optimized when the primer concentration was optimized as 0.24 μM. The RPA-LFD reaction was performed at six different temperatures: 30˚C, 35˚C, 37˚C, 39˚C, 45˚C, and 50˚C.

### Optimization of RPA-LFD reaction time

Next, the reaction time was optimized using 0.24 μM and 39˚C as the optimum concentration and temperature, respectively. RPA-LFD reactions were performed for seven different periods: 0 min, 5 min, 10 min, 15 min, 20 min, 25 min, and 30 min.

### Specificity test of RPA-LFD

For the specificity test, twenty different pathogenic bacteria were used (Table 1). Bacterial genomic DNA was extracted using the TIANamp Bacteria DNA Kit (TIANGEN BIOTECH Co., Ltd, Beijing, China) according the manufacturer's instructions, and DNA was quantified to the same concentration using NanoDrop 2000 (Thermo Fisher Scientific Co., Ltd, Shangha China). They were tested using the developed RPA-LFD assay specific for *ipaH* gene at the optimal concentration, temperature, and reaction time of 0.24 μM, 39˚C, and 20 min, respectively.

### Sensitivity determination of RPA-LFD

The recombinant plasmid puc57 containing an *ipaH* specific fragment was synthesized by Tsingke Biotechnology Co., Ltd.; the plasmid DNA was diluted ten folds in nuclease-free water to obtain a series of concentrations ranging from $1.29 \times 10^7$ copies/μL-$1.29 \times 10^0$ copies/μL. The plasmid solution was used as the reaction template for basic RPA and RPA-LFD and stored at -20˚C before use. The RPA-LFD reaction was carried out under the previously described optimal conditions.

### Detection of *Shigella* and EIEC in food sample

Sample of cucumber were used for realistic tests of the detection of *Shigella* and EIEC. The samples were prepared according to Zhang *et al.* [22]. The cucumber was homogenized after washing with ultrapure water and centrifuged at 10,000 rpm for 10 min to remove solid precipitates. The sample was then filtered using a 0.22 μm filter membrane to ensure sterility. Then, gradient dilutions of *Shigella* and EIEC were added to cucumber samples to obtain spiked samples with bacterial concentrations of $1.46 \times 10^2$ CFU/mL-$1.46 \times 10^5$ CFU/mL and $1.63 \times 10^2$ CFU/mL-$1.63 \times 10^5$ CFU/mL. The DNA extraction method of standard samples was the same as above. Then the RPA-LFD method was used to detect the standard samples and the results were compared with the traditional plate counting method.

## Results

### Optimal RPA primers

The designed RPA primer pairs are listed in S1 Table. According to the reaction results of basic RPA (S1 Fig), *ipaH* 03 was determined as the best primer pair.

### Optimal RPA-LFD probe

The primers and probes designed for *ipaH* 03 according to the RPA-LFD primer and probe design principles are listed in S2 Table. The *ipaH* probe 2 was determined to yield false positive results in the RPA-LFD assay so the *ipaH* probe 1 was selected as the assay probe (S2 Fig).

### Optimal primer concentration for RPA-LFD

As shown in Fig 1, the color of the LFD test line was lighter when the final concentrations of the primer were 0.08 μM and 0.16 μM. In comparison, when the final concentrations were 0.24 μM, 0.32 μM, and 0.40 μM, the color of the LFD test line was more obvious, whereas the color of the line did not significantly darken when the concentration exceeded 0.24 μM. Thus, 0.24 μM was ultimately selected as the optimal primer concentration.

### Optimal reaction temperature for RPA-LFD

The reaction temperature was optimized using a final primer concentration of 0.24 μM, as shown in Fig 2. When the reaction temperature was 30˚C, the color of the test line did not change. As reaction temperature increased, the color of the test line gradually became darker. There was no obvious color change at 35˚C or 37˚C, and the most intense at 39˚C. When the temperature reached 41˚C, the color intensity of the test line gradually decreased, and at 50˚C, no color change was observed. This phenomenon may be caused by the inactivation of enzymes in the system due to high temperature, so 39˚C was determined as the optimal reaction temperature.

### Optimal reaction time for RPA-LFD

Reaction time is also an important factor for RPA-LFD detection. The reaction time was optimized using a final primer concentration of 0.24 μM and reaction temperature of 39˚C, as shown in Fig 3. As time progressed, the color intensity of the LFD test line gradually increased. The color intensity was most intense at 20 min; no color intensification was observed after this point. Thus, we determined that 20 min was the optimal reaction time.

### Specificity of RPA-LFD

A total of 20 species of bacteria were used in the experiment to test the specificity of the RPA-LFD reaction under the optimal experimental conditions (Fig 4). The LFD results showed that, for all strains except *Shigella* spp. and EIEC, the LFD produced only control lines, which is indicative of a negative result. In contrast, the test line for *Shigella* and EIEC LFD showed a distinct red color. Therefore, the experimental results show that the developed RPA-LFD test has specificity for *ipaH*.

### Sensitivity determination of RPA-LFD

Experiments with the recombinant plasmid containing the *ipaH* gene were performed to evaluate the detection sensitivity of the RPA-LFD method, with RPA as the control. The comparative results of RPA-LFD and RPA with 10-fold-diluted recombinant plasmid ranging from

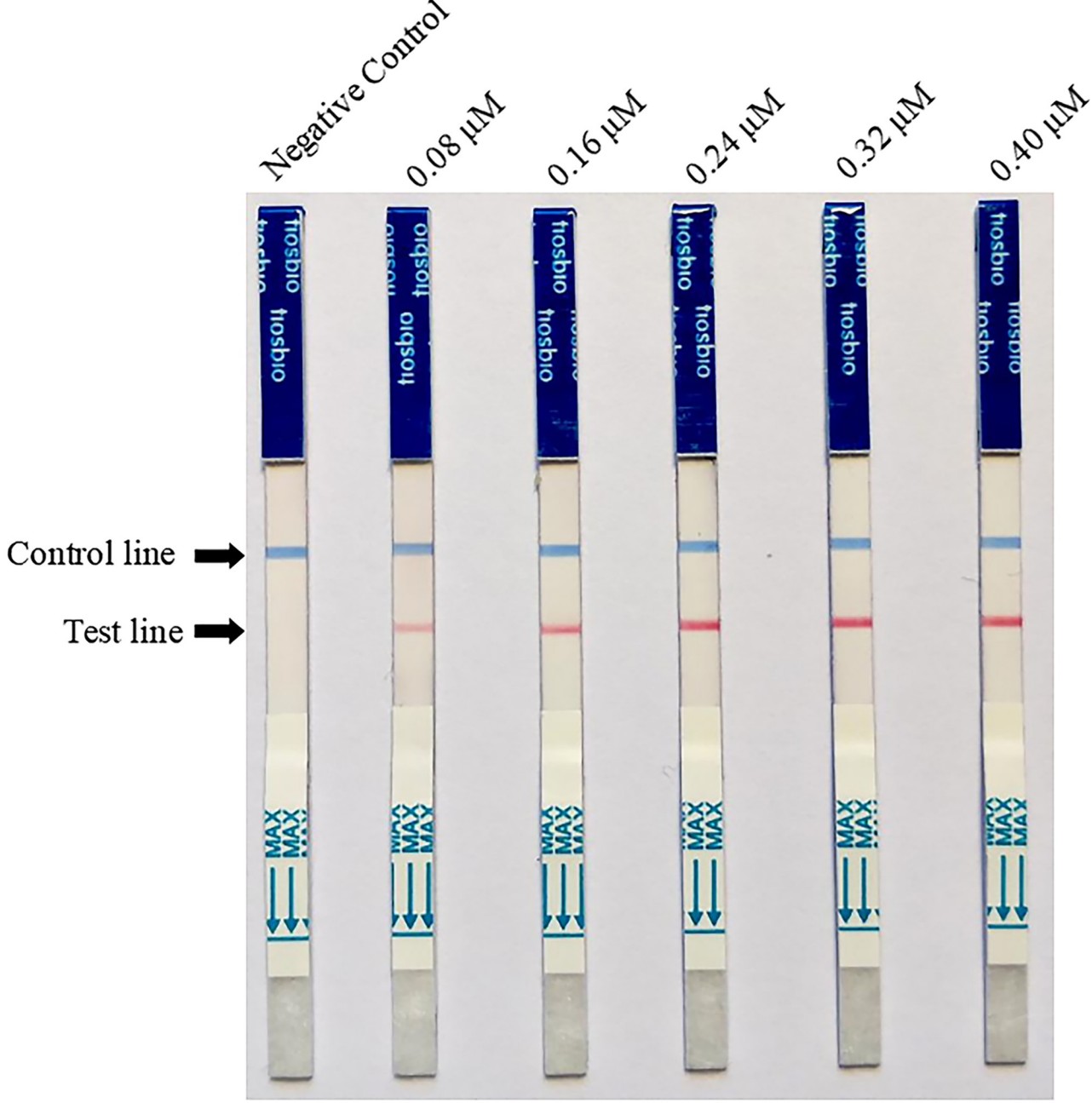

**Fig 1. Optimization of primer concentration for recombinase polymerase amplification combined with lateral flow test.** A darkened test line indicates a positive result.

$1.29 \times 10^7$ copies/µL-$1.29 \times 10^0$ copies/µL and nuclease-free water as the negative control are shown in Fig 5. The results showed that the lowest detection limit was the same for RPA-LFD and RPA, at $1.29 \times 10^2$ copies/µL. Although the sensitivity of RPA and RPA-LFD were the same, except for the 20 min of RPA reaction, LFD only took 3 min, while AGE took 30 min for detection.

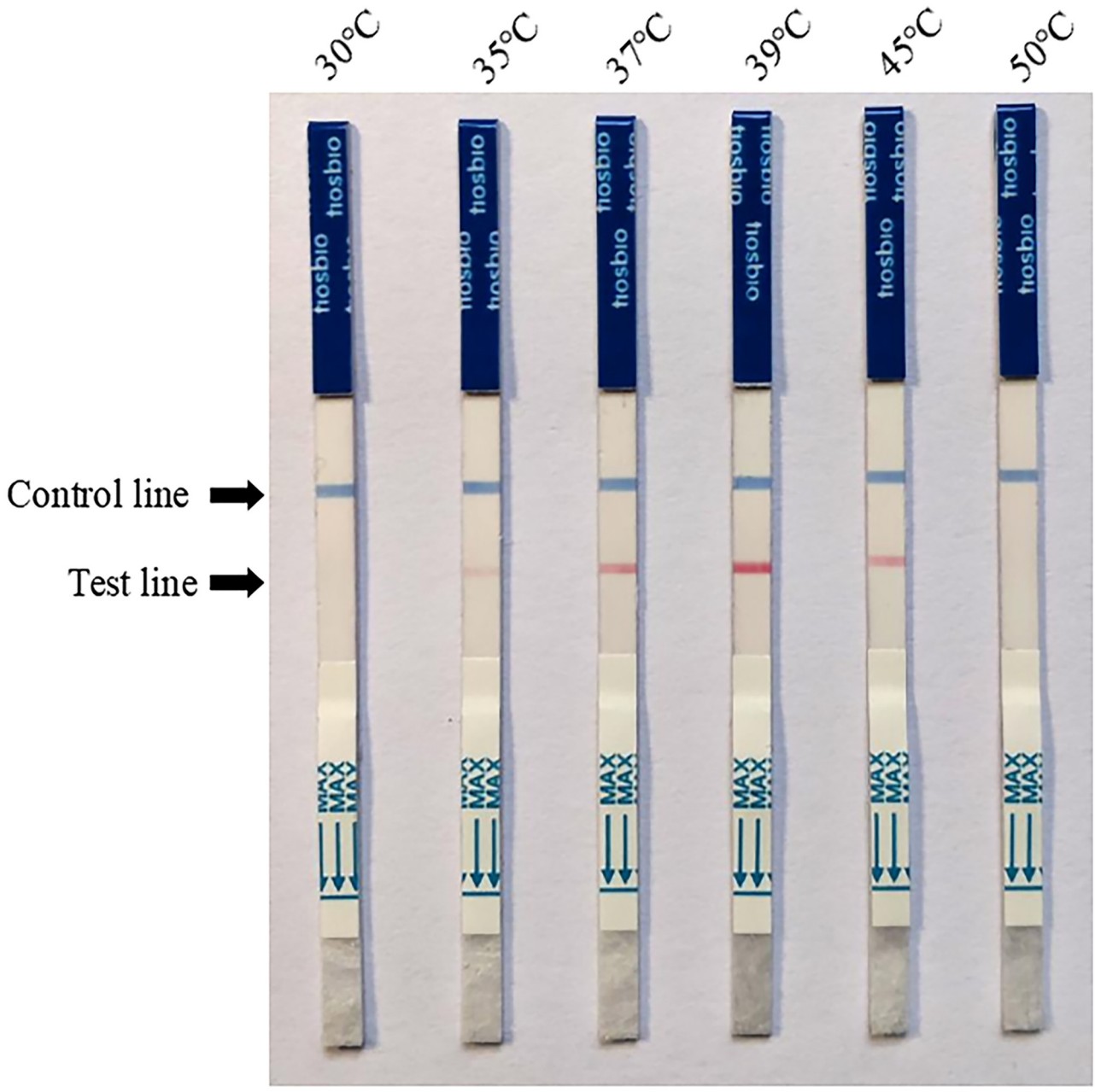

**Fig 2. Optimization of temperature for recombinase polymerase amplification combined with lateral flow test.** A darkened test line indicates a positive result.

## Food sample detection

To further investigate the application of RPA-LFD in the detection of *Shigella* and EIEC in actual sample, cucumber was used as a model and compared with the plate counting method. The analytical results are shown in Table 2. The detection limits of the RPA-LFD method for *Shigella* and EIEC in cucumber samples were $1.46 \times 10^3$ CFU/mL and $1.63 \times 10^3$ CFU/mL, respectively, which were slightly higher than the plate counting method. The results showed that the proposed RPA-LFD was feasible for the detection of *Shigella* and EIEC in actual samples.

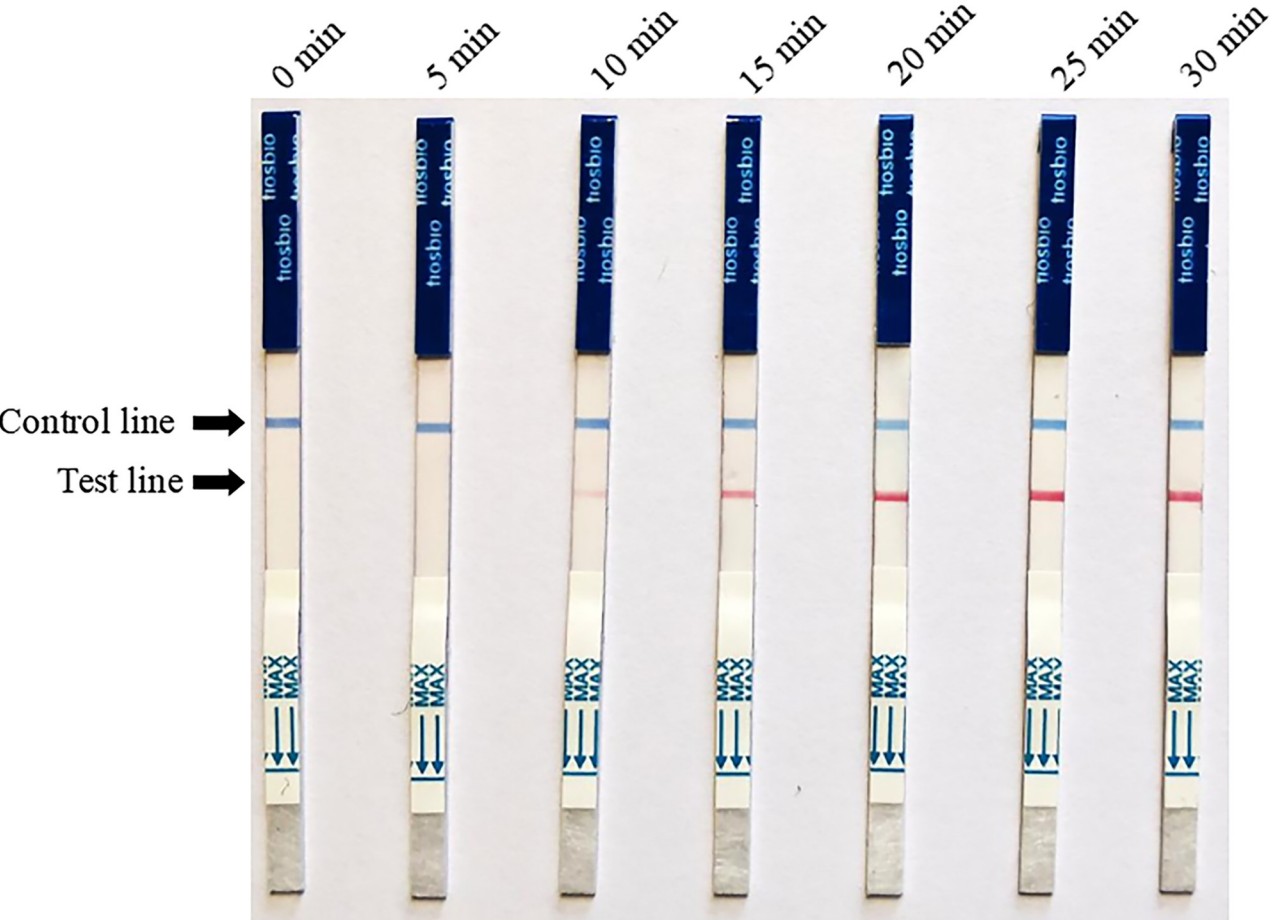

**Fig 3. Optimization of reaction time for recombinase polymerase amplification combined with lateral flow test.** A darkened test line indicates a positive result.

## Discussion

In this study, the RPA-LFD method was developed for rapid detection of *Shigella* and EIEC. In order to ensure the normal progress of the reaction, primers and probes with suitable sequence length and nucleotide composition were designed. The RPA-LFD detection method established in this study has dual specificity, namely primer specificity and probe specificity. The gene sequence encoding the invasive plasmid antigen H (*ipaH*) has been shown to be present on both *Shigella* and EIEC invasive plasmids and chromosomes, and is therefore often used as a target gene [23]. Therefore, in this study, we designed four different primer combinations for the *ipaH* gene and screened the most suitable primers for probe design and the amplification reaction. Designing an optimal probe further ensures the specificity of the RPA-LFD reaction, but the FAM-labeled probe easily combines with the biotin-labeled reverse primer to form a dimer and leads to false positive detection results. Therefore, we designed multiple sets of probes to use with the best primer pair to eliminate false positives and to ensure the accuracy of the results.

In addition to rapidness, specificity, and cost, sensitivity of the detection method was also tested. The detection method established in this study is comparable to other detection methods. For example, Song *et al.* [24] developed a new LAMP method to detect *Shigella*

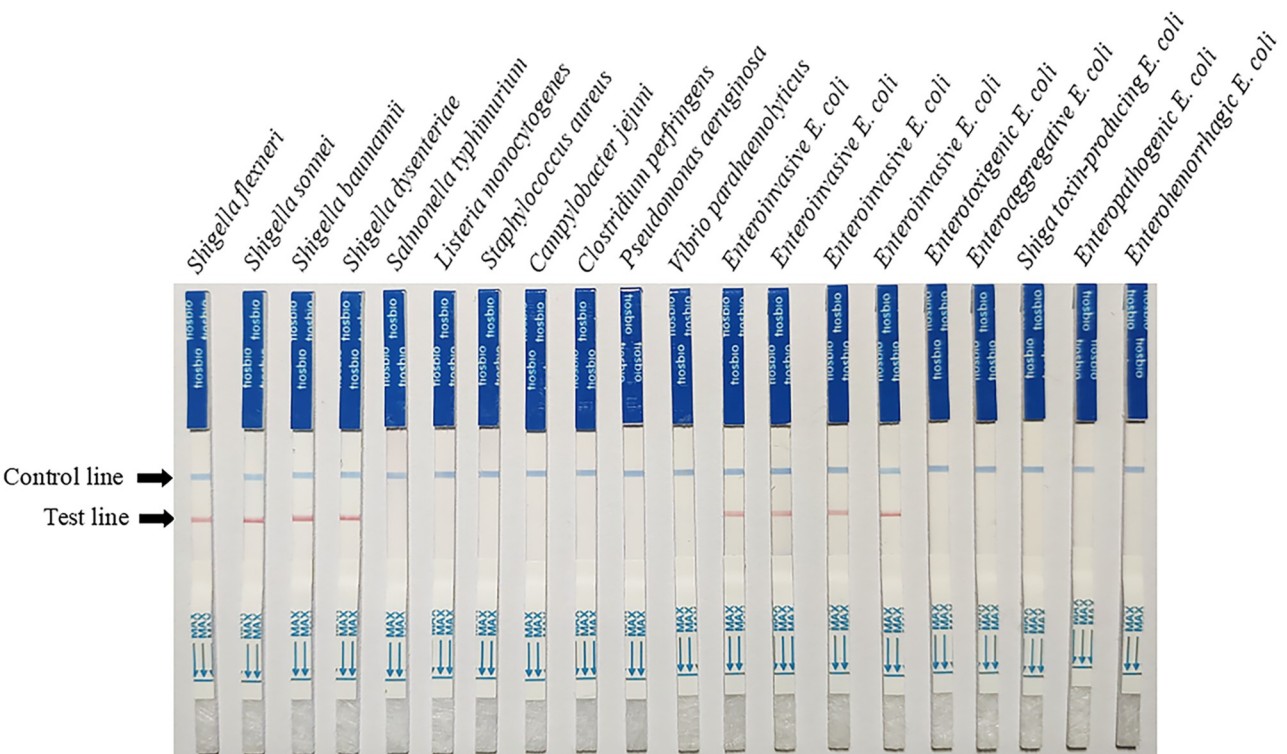

**Fig 4. Specificity test for recombinase polymerase amplification combined with lateral flow test.** *Shigella* and EIEC (the target strain) was tested against twelve different bacterial strains. A darkened test line indicates a positive result.

and enteroinvasive *Escherichia coli*, and the detection limit was 8 CFU per reaction. Liew *et al.* also used this method to detect with a detection limit of $5.9 \times 10^5$ CFU/mL [25]. Chen *et al.* developed a PCR and fluorescent microsphere (FM) immunoassay based on magnetic purification [26]. A chromatography test strip (ICTS) combined method has been used to detect *Shigella*, and the detection limit was $2.5 \times 10^{-7}$ ng/μL. Lukman *et al.* used a fast gold nanoparticle lateral flow analyzer to detect *Shigella* and *Salmonella* with a detection limit of $3.0 \times 10^6$ CFU/mL [27]. The detection limit in this study is significantly higher than that of the PCR detection method developed by Chandra *et al.*, Zhang *et al.*, and Barletta *et al.* [28–30].

In addition, we compared the sensitivities of RPA and RPA-LFD methods. The detection limits of both methods are similar at $10^2$ copies/μL. However, the entire RPA-LFD reaction only takes 25–30 min at 39°C for detection and does not require expensive or large instruments. In contrast, the RPA reaction requires agarose gel electrophoresis, which not only takes longer, but also requires bulky instruments, such as gel imagers. The RPA-LFD method established in this study is also comparable to other RPA-LFD used for the detection of pathogenic bacteria. The detection limit in this study is comparable to that of the RPA-LFD method developed by Hu *et al.* for detecting *Salmonella typhimurium* in milk and of the RPA-LFD method developed by Gao *et al.* for detecting *Salmonella* in shellfish [31, 32]. Furthermore, RPA-LFD only requires a constant temperature environment and the test LFD can be stored for several days. We also optimized the primer concentration for this reaction. So far, few studies have optimized the primer concentration for the RPA-LFD reaction, despite the fact that it has a direct effect on LFD results and that optimization can make the

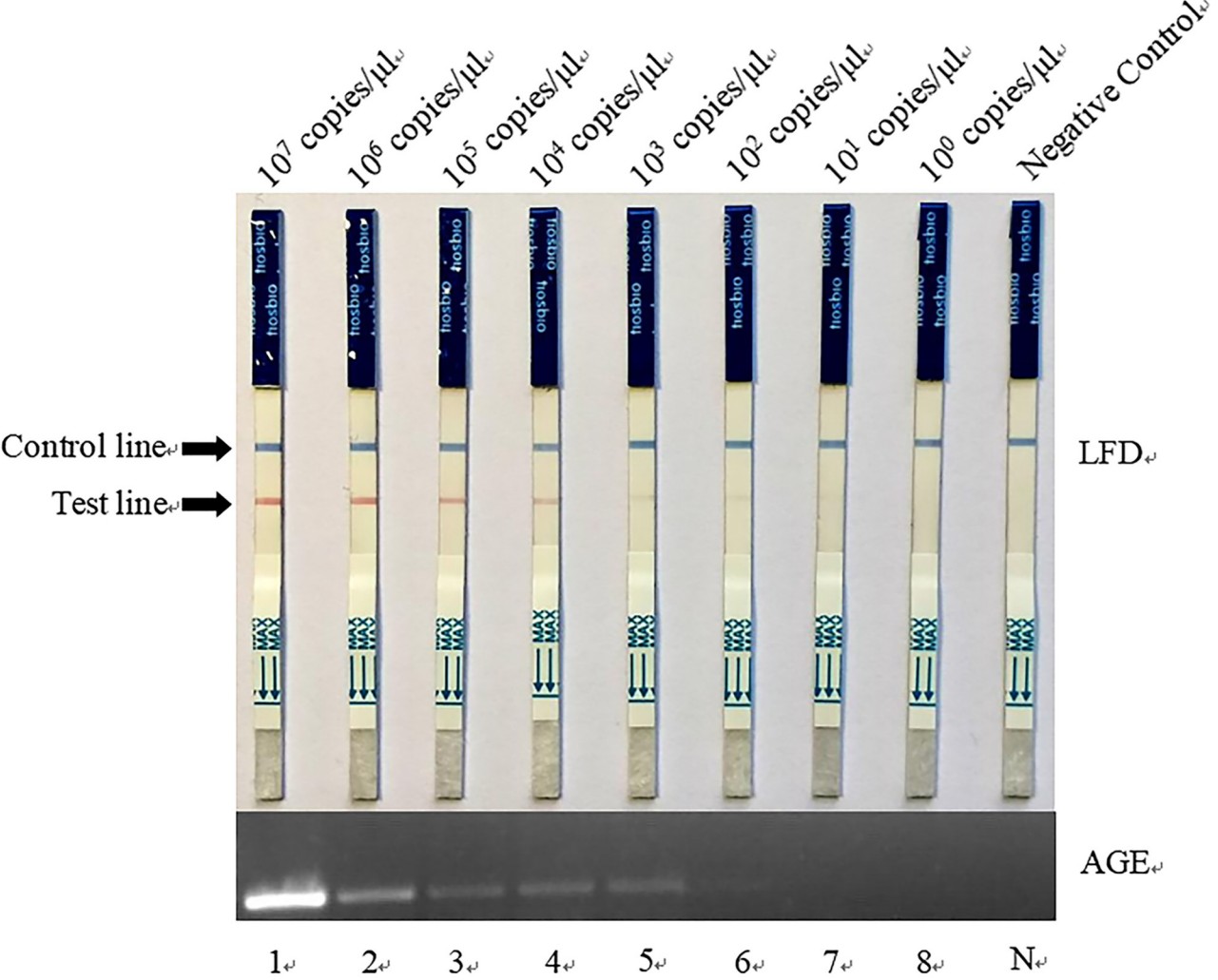

**Fig 5. Sensitivity of recombinase polymerase amplification combined with lateral flow test compared to that of recombinase polymerase amplification combined with agarose gel electrophoresis (RPA-AGE).** A standard 10-fold diluted plasmid was used. For RPA-AGE, 1–7 represent the results of the $10^7$–$10^0$ copies/μL dilutions, respectively. N: Negative control. A darkened test line indicates a positive result.

experiment more economical. At the same time, it is not difficult to find that the efficiency of DNA extraction has a certain impact on the actual detection through the actual sample detection. Therefore, the improvement of DNA extraction method is expected to further improve the sensitivity of RPA-LFD.

**Table 2. Detection of *Shigella* and EIEC in food sample.**

| Cucumber sample | Concentration of bacteria (CFU/mL) | | Our method | | Plate counting method | |
|---|---|---|---|---|---|---|
| | *Shigella* | EIEC | *Shigella* | EIEC | *Shigella* | EIEC |
| 1 | $1.46\times10^5$ | $1.63\times10^5$ | + [a] | + | + | + |
| 2 | $1.46\times10^4$ | $1.63\times10^4$ | + | + | + | + |
| 3 | $1.46\times10^3$ | $1.63\times10^3$ | + | + | + | + |
| 4 | $1.46\times10^2$ | $1.63\times10^2$ | - | - | + | + |

[a] The symbol "+" indicates that the method can detect bacteria; the symbol "-" indicates that the method cannot detect bacteria.

Nevertheless, the RPA-LFD method has some limitations. For example, the nonspecific amplification of primers and probes means that many repeated experiments may be needed to avoid false positive results. In addition, the popularity of the technology is not high, and thus LFD test strips are more expensive than AGE testing. Moreover, the basic LFD method cannot detect two or more nucleotide amplifications simultaneously in one RPA system, which makes the high-throughput detection of food-borne pathogens difficult. In order to solve this problem, RPA should be combined with multiple lateral flow test strips in future studies to achieve high-throughput detection. Furthermore, obtaining a positive result using this method is subjective, as it is qualitative and judged by the naked eye. Differences in eyesight and judgment between individuals may make it difficult to obtain accurate experimental results, and thus this methodology may need to be used in conjunction with an LFD reading instrument.

## Supporting information

**S1 Fig. Screening of primers for RPA reaction by agarose gel electrophoresis.** M: DNA marker. N: Negative control. 1–4: *ipaH* 01, *ipaH* 02, *ipaH* 03, *ipaH* 04. Under the same reaction conditions, the results of 3 agarose gel electrophoresis are more obvious.
(TIF)

**S2 Fig. Screening of recombinase polymerase amplification combined with lateral flow test probes.** 1: *ipaH* probe1, 2: *ipaH* probe2, N1: *ipaH* probe1 negative control, N2: *ipaH* probe2 negative control. Probe 1 showed a positive result. In negative control 1, only the control line changed color, indicating it was a valid control. Probe 2 produced very obvious false positive results.
(TIF)

**S1 Table. Primers for basic recombinase polymerase amplification (RPA) of *ipaH*.**
(DOCX)

**S2 Table. Primers and probes used for recombinase polymerase amplification combined with lateral flow test for the detection of *ipaH*.**
(DOCX)

## Acknowledgments

We would like to thank Editage (www.editage.cn) for English language editing.

## Author Contributions

**Conceptualization:** Yuanhong Xie.

**Data curation:** Junhua Jin.

**Formal analysis:** Yanling Hao, Linshu Jiang.

**Funding acquisition:** Yuanhong Xie.

**Investigation:** Zheng Bian, Linshu Jiang.

**Methodology:** Zheng Bian, Wenbo Liu, Hongxing Zhang.

**Project administration:** Yuanhong Xie.

**Resources:** Yanling Hao, Yuanhong Xie.

**Software:** Zheng Bian, Wenbo Liu.

**Supervision:** Hongxing Zhang.

**Validation:** Yuanhong Xie, Hongxing Zhang.

**Visualization:** Junhua Jin.

**Writing – original draft:** Zheng Bian.

**Writing – review & editing:** Yuanhong Xie, Hongxing Zhang.

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
