## [Decision Letter · Decision Letter 0]

30 Sep 2022

PONE-D-22-22928Development of a recombinase polymerase amplification assay with lateral flow dipstick (RPA-LFD) for rapid detection of the bacterium * Shigella*spp. and enteroinvasive * Escherichia coli*PLOS ONE

Dear Dr. Zhang,

Thank you for submitting your manuscript to PLOS ONE. After careful consideration, we feel that it has merit but does not fully meet PLOS ONE’s publication criteria as it currently stands. Therefore, we invite you to submit a revised version of the manuscript that addresses the points raised during the review process.

Please provide clarifications and revise addressing reviwer comments. 

We look forward to receiving your revised manuscript.

Kind regards,

Iddya Karunasagar

Academic Editor

PLOS ONE

Journal Requirements:

"We thank the Research project of Beijing Municipal Commission of Education, this work was supported by the Research project of Beijing Municipal Commission of Education (KM201810020016)."

Additional Editor Comments:

Please see the reviewer comments and provide clarifications and revision addressing all comments point by point.

Reviewers' comments:

Reviewer's Responses to Questions

**Comments to the Author**

1. Is the manuscript technically sound, and do the data support the conclusions?

Reviewer #1: Yes

Reviewer #2: Partly

2. Has the statistical analysis been performed appropriately and rigorously? 

Reviewer #1: N/A

Reviewer #2: N/A

3. Have the authors made all data underlying the findings in their manuscript fully available?

Reviewer #1: Yes

Reviewer #2: Yes

4. Is the manuscript presented in an intelligible fashion and written in standard English?

Reviewer #1: No

Reviewer #2: Yes

5. Review Comments to the Author

Reviewer #1: The authors have developed an RPA-LFD for rapid detection of Shigella spp. and enteroinvasive Escherichia coli using bacterial pure culture.

The optimized assay cannot differentiate Shigella spp. and enteroinvasive Escherichia coli, and they cross-reacted with each other; Hence the title should be revised accordingly. Also, this can be discussed in the discussion section.

What was the rationale for using the ipaH gene?

Line: 81-82: mention the details of the laboratory from where bacterial strains were procured

Table 1: typhimurium can be written as “Typhimurium” and non-italicize

Line 101: mention a reference for this “phenol-chloroform method”

Reviewer #2: PONE-D-22-22928

Development of a recombinase polymerase amplification assay with lateral flow dipstick (RPA-LFD) for rapid detection of the bacterium Shigella spp. and enteroinvasive Escherichia coli

This study developed a recombinase polymerase-based assay followed by lateral flow dipstick (RPA-LFD) detection of Shigella spp. and the enteroinvasive E. coli. The ipaH gene common to both the organisms was used as the target for RPA. This naked eye detection method is rapid and specific to the target organisms.

Major comments:

Although the study successfully developed RPA-LFD method, its utility has not been demonstrated beyond identifying pure cultures of bacteria. Precisely, where this assay can be employed? For clinical specimens, foods or water?. The performance of the assay can vary in different matrices. In this context, although the study has accomplished developing a RPA-LFD method for Shigella and EIEC, it has not demonstrated it applicability to any sample type. Apart from the sample matrix, the DNA extraction method also has bearing on the sensitivity of the assay.

The specificity of the assay can also vary depending on the background microbiota, which again can vary across the the sample types. It is necessary that the specificity of the assay is established using the natural samples.

The sensitivity of RPA-LFT was determined using the plasmid containing cloned fragment of ipaH gene. Ideally, this should have been done using 10-fold serially diluted bacterial cultures (Shigella and EIEC), followed by DNA extraction.

Minor comments

Title: Consider removing “the bacterium” from the title

L40: “pathogens”.

L45-46: Please provide references to this statement.

L50: Please replace “abundant” with “present”

Table 1: Please expand CICC in the foot note, and the source/origin

L95: Please clarify what did you mean by “NCBI was used to design primers”

L99-100: “After mixing, 2.5 µL of Mg(Ac)2 (280mM) was added to start the reaction”.

L109: Please delete “Most of..”

L178-180: Please rephrase this sentence to convey the correct interpretation

6. PLOS authors have the option to publish the peer review history of their article (what does this mean?). If published, this will include your full peer review and any attached files.

Reviewer #1: No

Reviewer #2: No

---

## [Author Response · Author response to Decision Letter 0]

18 Oct 2022

Reviewer: 1

Q1: The optimized assay cannot differentiate Shigella spp. and enteroinvasive Escherichia coli, and they cross-reacted with each other; Hence the title should be revised accordingly. Also, this can be discused in the discussion section.

A: The title has been modified according to your suggestions and discussed in the discussion.

Q2: What was the rationale for using the ipaH gene?

A: We focused on foodborne pathogens that can produce toxins in food, and the ipaH gene is one of the major genes encoding virulence factors in Shigella and enteroinvasive Escherichia coli. In addition, Hartman et al. [1] found that ipaH gene has multi-copy nature and exists on both chromosome and invasion plasmid, which makes it superior to other single-copy virulence genes. At the same time, Lampel and Orlandi [2] reported that storage before testing would lead to the loss of invasive plasmids. Molecular assays designed to detect single-copy genes on invasive plasmids may fail, so molecular assays targeting ipaH genes have a greater chance to detect invasive plasmids or chromosomes.

Q3: Line: 81-82: mention the details of the laboratory from where bacterial strains were procured

A: The manuscript has been revised according to your suggestion.

Q4: Table 1: typhimurium can be written as “Typhimurium” and non-italicize

A: The manuscript has been revised according to your suggestion.

Q5: Line 101: mention a reference for this “phenol-chloroform method”

A: The manuscript has been revised according to your suggestion.

Reviewer: 2

Q1: The specificity of the assay can also vary depending on the background microbiota, which again can vary across the sample types. It is necessary that the specificity of the assay is established using the natural samples.

A: The actual sample testing experiment has been supplemented according to your suggestion.

Q2: The sensitivity of RPA-LFD was determined using the plasmid containing cloned fragment of ipaH gene. Ideally, this should have been done using 10-fold serially diluted bacterial cultures (Shigella and EIEC), followed by DNA extraction.

A: Because the extraction method of DNA sample will also affect the sensitivity of detection, we use standard plasmid for 10-fold gradient dilution in order to pursue the accuracy of detection limit. In the actual sample inspection, in order to achieve the universality of detection, use 10-fold of continuously diluted bacterial cultures (Shigella and EIEC) according to your recommendations, and then extract DNA.

Q3: Consider removing “the bacterium” from the title

A: The title has been revised according to your suggestions

Q4: L40: “pathogens”.

A: The manuscript has been revised according to your suggestion.

Q5: L45-46: Please provide references to this statement.

A: The manuscript has been revised according to your suggestion.

Q6: L50: Please replace “abundant” with “present”

A: The manuscript has been revised according to your suggestion.

Q7: Table 1: Please expand CICC in the foot note, and the source/origin

A: The manuscript has been revised according to your suggestion.

Q8: L95: Please clarify what did you mean by “NCBI was used to design primers”

A: Primers were designed and screened using the primer-BLAST function of the National Center for Biotechnology Information (NCBI).And the manuscript has been revised.

Q9: L99-100: “After mixing, 2.5 µL of Mg(Ac)2 (280mM) was added to start the reaction”.

A: The manuscript has been revised according to your suggestion.

Q10: L109: Please delete “Most of..”

A: The manuscript has been revised according to your suggestion.

Q11: L178-180: Please rephrase this sentence to convey the correct interpretation

A: The manuscript has been revised according to your suggestion.

Reference

1. Hartman, A. B., M. Venkatesan, E. V. Oaks, and J. M. Buysee. Sequence and molecular characterization of multicopy invasion plasmid antigen, ipaH of Shigella flexneri. J. Bacteriol. 1990; 172: 1905–1915.

2. Lampel, K. A., and P. A. Orlandi. Polymerase chain reaction detection of invasive Shigella and Salmonella enterica in food. Methods Mol. Biol. 2002; 179: 235–244.

---

## [Decision Letter · Decision Letter 1]

18 Nov 2022

PONE-D-22-22928R1Development of a recombinase polymerase amplification assay with lateral flow dipstick (RPA-LFD) for rapid detection of the *ipaH* gene from* Shigella*spp. and enteroinvasive * Escherichia coli*PLOS ONE

Dear Dr. Zhang,

Thank you for submitting your manuscript to PLOS ONE. After careful consideration, we feel that it has merit but does not fully meet PLOS ONE’s publication criteria as it currently stands. Therefore, we invite you to submit a revised version of the manuscript that addresses the points raised during the review process.

Please see the comment on title modification. 

We look forward to receiving your revised manuscript.

Kind regards,

Iddya Karunasagar

Academic Editor

PLOS ONE

Journal Requirements:

Additional Editor Comments:

Please see the reviewers comment regarding modification of title to better reflect the contents of the paper.

Reviewers' comments:

Reviewer's Responses to Questions

**Comments to the Author**

1. If the authors have adequately addressed your comments raised in a previous round of review and you feel that this manuscript is now acceptable for publication, you may indicate that here to bypass the “Comments to the Author” section, enter your conflict of interest statement in the “Confidential to Editor” section, and submit your "Accept" recommendation.

Reviewer #1: All comments have been addressed

Reviewer #2: All comments have been addressed

2. Is the manuscript technically sound, and do the data support the conclusions?

Reviewer #1: Yes

Reviewer #2: Yes

3. Has the statistical analysis been performed appropriately and rigorously? 

Reviewer #1: N/A

Reviewer #2: Yes

4. Have the authors made all data underlying the findings in their manuscript fully available?

Reviewer #1: Yes

Reviewer #2: Yes

5. Is the manuscript presented in an intelligible fashion and written in standard English?

Reviewer #1: Yes

Reviewer #2: Yes

6. Review Comments to the Author

Reviewer #1: (No Response)

Reviewer #2: The authors have modified the manuscript sufficiently as suggested in my first review.

However, the title might require modification. It says "....... detection of the ipaH gene from Shigella spp. and

enteroinvasive Escherichia coli". Here you goal is not just to detect ipaH gene. Instead, you are using this gene as a marker to detect two species of bacteria. Therefore, the title can be "Development of a recombinase polymerase amplification assay with lateral flow dipstick (RPA-LFD) for rapid detection of Shigella spp. and

4 enteroinvasive Escherichia coli

7. PLOS authors have the option to publish the peer review history of their article (what does this mean?). If published, this will include your full peer review and any attached files.

Reviewer #1: No

Reviewer #2: No

---

## [Author Response · Author response to Decision Letter 1]

19 Nov 2022

Dear reviewer:

Thank you very much for your comments. We have revised our manuscript according to your suggestions. And the detail information is following.

Sincerely,

Yuanhong Xie

Reviewer: 2

Q1: However, the title might require modification. It says "....... detection of the ipaH gene from Shigella spp. and enteroinvasive Escherichia coli". Here you goal is not just to detect ipaH gene. Instead, you are using this gene as a marker to detect two species of bacteria. Therefore, the title can be "Development of a recombinase polymerase amplification assay with lateral flow dipstick (RPA-LFD) for rapid detection of Shigella spp. and enteroinvasive Escherichia coli

A: The title has been modified according to your suggestions.

---

## [Editor Report · Decision Letter 2]

28 Nov 2022

Development of a recombinase polymerase amplification assay with lateral flow dipstick (RPA-LFD) for rapid detection of * Shigella*spp. and enteroinvasive * Escherichia coli*

PONE-D-22-22928R2

Dear Dr. Zhang,

We’re pleased to inform you that your manuscript has been judged scientifically suitable for publication and will be formally accepted for publication once it meets all outstanding technical requirements.

Kind regards,

Iddya Karunasagar

Academic Editor

PLOS ONE

Additional Editor Comments (optional):

All reviewer comments have been addressed.
---

## [Editor Report · Acceptance letter]

1 Dec 2022

PONE-D-22-22928R2 

Development of a recombinase polymerase amplification assay with lateral flow dipstick (RPA-LFD) for rapid detection of *Shigella* spp. and enteroinvasive *Escherichia coli*

Dear Dr. Zhang:

I'm pleased to inform you that your manuscript has been deemed suitable for publication in PLOS ONE. Congratulations! Your manuscript is now with our production department. 

Kind regards, 

on behalf of

Dr. Iddya Karunasagar 

Academic Editor

PLOS ONE